# Depiction of Branched-Chain Amino Acids (BCAAs) in Diabetes with a Focus on Diabetic Microvascular Complications

**DOI:** 10.3390/jcm12186053

**Published:** 2023-09-19

**Authors:** Daniela Maria Tanase, Evelina Maria Gosav, Tina Botoc, Mariana Floria, Claudia Cristina Tarniceriu, Minela Aida Maranduca, Anca Haisan, Andrei Ionut Cucu, Ciprian Rezus, Claudia Florida Costea

**Affiliations:** 1Department of Internal Medicine, “Grigore T. Popa” University of Medicine and Pharmacy, 700115 Iasi, Romania; tanasedm@gmail.com (D.M.T.); dr.evelinagosav@gmail.com (E.M.G.); 2Internal Medicine Clinic, “St. Spiridon” County Clinical Emergency Hospital Iasi, 700111 Iasi, Romania; minela.maranduca@umfiasi.ro; 3Department of Ophthalmology, Faculty of Medicine, “Grigore T. Popa” University of Medicine and Pharmacy, 700115 Iasi, Romania; tina.botoc@gmail.com (T.B.); claudia.costea@umfiasi.com (C.F.C.); 42nd Ophthalmology Clinic, “Prof. Dr. Nicolae Oblu” Emergency Clinical Hospital, 700309 Iasi, Romania; 5Department of Morpho-Functional Sciences I, Discipline of Anatomy, “Grigore T. Popa” University of Medicine and Pharmacy, 700115 Iasi, Romania; cristinaghib@yahoo.com; 6Hematology Clinic, “St. Spiridon” County Clinical Emergency Hospital, 700111 Iasi, Romania; 7Department of Morpho-Functional Sciences II, Discipline of Physiology, “Grigore T. Popa” University of Medicine and Pharmacy, 700115 Iasi, Romania; 8Department of Emergency Medicine, “Grigore T. Popa” University of Medicine and Pharmacy, 700115 Iasi, Romania; anca.haisan@umfiasi.ro; 9Emergency Department, “St. Spiridon” County Clinical Emergency Hospital, 700111 Iasi, Romania; 10Department of Biomedical Sciences, Faculty of Medicine and Biological Sciences, “Ștefan cel Mare” University, 720229 Suceava, Romania; andrei.cucu@usm.ro; 11Department of Neurosurgery, “Prof. Dr. Nicolae Oblu” Emergency Clinical Hospital, 700309 Iasi, Romania

**Keywords:** diabetes mellitus, T2DM, branched chain amino acids, BCAAs, diabetic retinopathy, diabetic nephropathy

## Abstract

Type 2 diabetes mellitus (T2DM) still holds the title as one of the most debilitating chronic diseases with rising prevalence and incidence, including its complications such as retinal, renal, and peripheral nerve disease. In order to develop novel molecules for diagnosis and treatment, a deep understanding of the complex molecular pathways is imperative. Currently, the existing agents for T2DM treatment target only blood glucose levels. Over the past decades, specific building blocks of proteins—branched-chain amino acids (BCAAs) including leucine, isoleucine, and valine—have gained attention because they are linked with insulin resistance, pre-diabetes, and diabetes development. In this review, we discuss the hypothetical link between BCAA metabolism, insulin resistance, T2DM, and its microvascular complications including diabetic retinopathy and diabetic nephropathy. Further research on these amino acids and their derivates may eventually pave the way to novel biomarkers or therapeutic concepts for the treatment of diabetes and its accompanied complications.

## 1. Introduction

Type 2 diabetes mellitus (T2DM) is one of the most prevalent chronic diseases worldwide, which is also recognized as a major threat to people’s health. The World Health Organization (WHO) estimates that about 422 million people globally have diabetes, and more than 95% of them have T2DM [1]. T2DM development is induced by two main factors: pancreatic β-cells dysfunction and peripheral insulin resistance (IR). The pancreas, liver, skeletal muscle, kidneys, brain, small intestine, and adipose tissue all partake in T2DM progression through important pathophysiological processes such as adipokine dysfunction, oxidative stress, inflammation, and abnormalities in gut microbiota [2].

In the last years, substantial efforts and technological advances brought novel diagnoses and therapeutic tools in diabetes management with current up-to-date practical guidelines; however, T2DM continues to remain one of the most debilitating diseases with multiple cardiovascular, renal, and neuronal complications [3]. Accordingly, the scientific community brings attention the amino acids, as organic compounds with valuable properties that ensure optimal health [3].

For over a decade, these metabolites have been used not only as biomarkers but also, as potential drug therapy targets. Among the gamut of amino acids, notably, essential branched-chain amino acids (BCAAs) were found to be connected to T2DM development and evolution [4]. They play a substantial role in protein synthesis enhancement in the muscles during physical stress conditions; are key regulators in nutrient sensing, neurotransmitter synthesis, and cellular signaling; produce energy support; and can perform as nitrogen donors. Accordingly, BCAAs have an intriguing role in microvascular diabetic complications such as diabetic retinopathy (DR), neuropathy (DN), and diabetic nephropathy [1,5], which renders their important contribution and thus encourages further intensive research in this field.

In this descriptive review, we discuss the hypothetical link among BCAA metabolism, insulin resistance, T2DM, and its microvascular complications diabetic retinopathy, respectively diabetic nephropathy. Furthermore, the potential utility of BCAAs as T2DM novel biomarkers and their potential in T2DM treatment are highlighted.

## 2. BCAA Catabolism and Metabolism

The BCAAs, namely, valine, leucine, and isoleucine, represent three of the nine essential amino acids that cannot be synthesized by the human body. For this reason, they must be obtained from the diet [6]. The most common dietary sources of BCAAs are high-fat dairy food items, red meat, and poultry, as well as synthetic supplements [6]. Usually, BCAA supplementation or BCAA-rich diets have beneficial effects on the regulation of homeostasis; however, they have been associated with obesity, IR, and increased risk of T2DM in the general population [7] and even in patients with a history of gestational diabetes [8].

Their catabolic journey is complex: once BCAAs reach cells, they are converted to their respective branched-chain α-keto acids (BCKAs), 2-ketoisocaproate (KIC), 2-keto-3-methylvalerate (KMV), 2-ketoisovalerate (KIV) from leucine, isoleucine, and valine [9]. This reversible reaction is catalyzed by branched-chain aminotransferases (BCATs) that exist in two isoforms: the cytosolic (BCATc or BCAT1) form and the mitochondrial (BCATm or BCAT2) form. BCAT1 is mainly expressed in neuronal tissues and immune cells, such as activated T lymphocytes and macrophages, whilst BCAT2 can be found in the majority of tissues with the exception of the liver; as a result, the first catabolic site of BCAAs is the skeletal muscles, not the liver as in other amino acids cases [9,10]. Next comes the irreversible oxidative decarboxylation of the α-ketoacids, catalyzed by the branched-chain α-ketoacid dehydrogenase (BCKD), which consists of three catalytic components: E1, E2, and E3. By this process, the carbon skeletons of KIC, KMV, and KIV are converted to their respective ketoacids: isovaleryl-CoA, 2-methylbutyryl-CoA, and isobutyryl-CoA [7,11]. The activity of the BCKD complex is inhibited by the phosphorylation of the E1a unit induced by BCKD kinase and by an increased NADH/NAD+ rate, while its dephosphorylation, induced by protein phosphatase 2Cm (PP2Cm), leads to BCKD activation [7,9].

The last step in BCAA catabolism leads to adenosine triphosphate (ATP) production separately for each amino acid. KIC catabolism leads to acetyl-CoA and acetoacetate production, while KIV is transformed into succinyl-CoA, all leading to the tricarboxylic acid cycle (TCA), which is also known as the Krebs cycle or the citric acid cycle [9,12]. These catabolic complex and intricate pathways may hold therapeutic value and are worth future investigations. Additionally, BCAAs induce metabolic and signaling functions by activating the mechanistic target of rapamycin (mTOR) signaling pathway. mTOR is a cytoplasmic serine/threonine protein kinase that consists of two complexes: mTORC1 and mTORC2 [13].

Out of all BCAAs, leucine is the strongest mTORC1 activator [14]. It induces mTORC1 activation by binding Sestrin2, a negative regulator of mTORC1 activity. In the absence of leucine, Sestrin2 inhibits GATOR2 (a receptor for the cytosolic amino acid sensors), a positive regulator of mTORC1 activity. When leucine is available at high concentrations, Sestrin2 releases GATOR2, leading to mTORC1 activation [14]. When mTORC1 is activated by excessive glucose or branched chain amino acid consumption, the insulin receptor substrate phosphorylation increases. This alters its insulin sensitivity, thereby rendering the cells in an insulin-resistant state. As a result, myocytes, adipocytes, hepatocytes, and pancreatic β-cells develop an impaired glucose tolerance, which may lead to type 2 diabetes [15].

Another hypothesis would be that BCAA dysmetabolism leads to toxic metabolite accumulation, which induces elevated reactive oxygen species (ROS) levels and mitochondrial dysfunction in the brain, liver, heart, and pancreatic β cells and, ergo, insulin resistance and T2DM promotion [16]. Moreover, excessive diet supplementation with BCAAs seems to also activate mTORC1 signaling, which may lead to IR and T2DM (Figure 1) [17]. Also, Zhang et al. [17] showed that BCAAs provoke hepatic damage in high-fat diet mice, involving both adipocytes and hepatic cells, evidenced by hepatic oxidative stress, hepatic apoptosis, and increased levels of circulating liver enzymes.

### 2.1. Insulin Resistance and Inflammation

Increased circulating BCAA levels and/or their derived intermediates, BCKAs, in individuals with obesity and insulin resistance were first reported nearly a half-century ago [18]; however, their role in the future development of these conditions is not still clear. Allam-Ndoul et al. [19] showed that plasma BCAA concentrations might serve as a better indicator of IR in a pre-diabetic state than plasma glucose levels. Jang et al. [20] identified that 3-hydroxyisobutyrate (3-HIB) is elevated in diabetic individuals, a paracrine regulator of trans-endothelial fatty acid transport in the skeletal muscle in mice, and can lead to increased esterified lipid accumulation, lipotoxicity, impaired insulin signaling, and future development of IR.

Moghei et al. [21] proved that KIC, an obligatory metabolite of leucine, stimulated mTORC1 signaling but suppressed insulin-stimulated glucose transport in the absence of other amino acids, further increasing the demand for insulin. Along with inflammation and lipotoxicity associated with IR, this led to hyperinsulinemia and pancreatic beta cell malfunctioning, suggesting its progressive transition to T2DM. In addition, BCAA deprivation in mice increases pAKT hepatic levels (a marker of insulin signaling) and improves insulin sensitivity [18]. Thus, BCAAs not only have a correlative, but also a causal role in the development of obesity, IR, and diabetes.

BCAA catabolism is significantly lower in the adipose tissue of obese and insulin-resistant patients because of the lower expression and activity of BCAT2 and BCKD induced by the phosphorylation and resultant suppression of the E1 subunit of the BCKD complex. Slowed BCAA catabolism in adipose tissue negatively influences its differentiation and leads to excess lipid accumulation in adipocytes and further IR [9]. Additionally, fructose-rich intakes enhance Ch REBP-β factor transcription function, which leads to an elevated BCKDK: PP2Cm ratio and low BCKD complex expression. Moreover, it takes part in the phosphorylation and activation of the ATP citrate lyase (ACL), which stimulates the formation of lipogenic substrates, elevated fatty acid synthesis, and dyslipidemia, altogether leading to IR development [9].

Increased BCAAs production by gut microbiota and decreased expression of BCAAs catabolizing enzymes in white adipose tissue (WAT) is considered to increase serum levels of BCAAs in states of insulin resistance and obesity, causing incomplete lipid oxidation in muscle with acyl carnitines and mitochondrial dysfunction [22]. The branched-chain α-ketoacid dehydrogenase complex (BCKDH) has a lower expression in skeletal muscle, so it is more difficult for it to decompose BCKAs in comparison with the liver, where BCKAs decompose in large amounts, leading to the accumulation of multiple acyl carnitines, which damages the mitochondrial TCA cycle and leads to insulin resistance [23].

In a similar manner, high intakes of BCAAs may lead to IR development [24]. It has been found that disturbed amino acid (AA) dysmetabolism, revealed during the early postpartum period, precedes the progression to T2DM among women with gestational diabetes mellitus (GDM) [24]. Some mild disturbances have been established in GDM cases compared with T2DM cases, where an entire pathway was dysfunctional [21]. Importantly, metabolites like 1-carboxyethylleucine, 1-carboxyethylvaline, and 1-carboxyethylisoleucine, which were significantly higher in those with GDM or diabetes, were lower in breastfeeding women [25]. Also, patients with GDM who follow metformin treatment present a higher rate of dysregulated BCAA metabolism compared with patients who follow insulin therapy [26]. Because weight loss with caloric restriction and reduced BCAA intake are prohibited in pregnancies, a persistent elevation of BCAAs in GDM patients leads to dysregulated energy metabolism and is highly predictive for GDM to T2DM transition [27].

Inflammation is one of the key participants identified in many metabolic diseases including diabetes, obesity, and cardiovascular diseases. A high-fat diet promotes free fatty acid accumulation, which generates gut dysbiosis, and the enhancement of pro-inflammatory cell production (cytokines such as IL-6 and chemokines) with subsequent redox imbalance and oxidative stress. In this matter, Liu et al. [28] noted that a long-term high BCAA supplementation further increased BCKA levels, inflammation, and tissue fibrosis (liver and kidney) in rodents. Interestingly, women with higher circulating BCAA concentrations had higher lipoprotein insulin resistance scores and levels of inflammatory and dyslipidemia biomarkers independent of established cardiovascular disease risk factors [29].

In the last decade, inflammatory markers such as IL-6, fibrinogen, and hs-CRP rendered an independent causative role for subclinical inflammation, demonstrating their involvement in diabetes and its related complications. BCAAs and different types of fatty acids regulate the inflammatory response through the NF-κB pathway and NLRP3. Researchers also found a significant relationship between isoleucine and IL-6 in patients with metabolic syndrome, suggesting isoleucine is a potential predictive biomarker of the pro-inflammatory state of nascent metabolic syndrome [30]. A recent clinical trial (NCT02613741) demonstrated that compared with normal weight, the concentration of BCAAs exhibited significant statistical baseline correlations with body composition and inflammatory factors [31]. These novel findings of the BCAA-inflammation relationship may raise the prospect of BCAA as a biomarker and evoke a potential link between obesity, T2DM, and CVD.

### 2.2. BCAA Metabolic Gene Expression in Diabetes

Currently, the majority of metabolomic studies are limited to an analysis of plasma metabolites under fasting conditions. The transcriptomic regulation of genes involved in BCAA catabolism/metabolism is incompletely unknown, and the metabolomic profiles of peripheral tissues involved in glucose homeostasis are insufficient. In this regard, after a glucose load under fasting conditions, alterations in BCAA catabolism in patients with T2DM were observed, highlighting metabolic inflexibility. These changes were accompanied by a stronger correlation with downregulated expression of branched-chain keto acid dehydrogenase complex subunits and downstream BCAA-related genes in skeletal muscle. Transcriptional regulation of BCAA genes in primary human myotubes through peroxisome proliferator-activated receptor γ coactivator-1α (PGC-1α) is ERRα-dependent [32]. It seems that higher BCAA intakes exert an unfavorable effect on T2DM risk only among those with high genetic susceptibility and that a genetic predisposition to BCAA metabolism disorder alters the effect of dietary BCAA intakes on T2DM [33].

Jang et al. [20] found that patients with early-onset T2DM displayed 449 dysregulated genes: 268 upregulated, while patients with classical-onset T2D exhibited 790 dysregulated genes: 366 upregulated and 424 downregulated. The higher number of dysregulated genes detected in patients with the classical debut of the disease implies that gene dysregulation might be influenced by the disease’s time span. Additionally, patients with classical T2DM showed reduced expression of BCAT2 and BCKDHB, and the same subjects with early-onset T2D exhibited reduced expression of skeletal muscle BCAT2. These results emphasize that the genes regulating the expression of BCAAs in skeletal muscles are expressed differently depending on T2DM onset.

## 3. Diabetic Retinopathy

Diabetic retinopathy (DR) is one of the most common microvascular complications of diabetes that has risen to alarming levels, as it is identified in a third of diabetic patients [34,35]. Chronic hyperglycemia reduces retinal vascular endothelial functions, causes retinal ischemia, and increases vascular permeability; DR is considered a leading cause of vision loss [36]. DR can be classified as non-proliferative diabetic retinopathy (NPDR) with increased vascular permeability, basal membrane thickening, and loss of pericytes in the retinal capillaries. It usually progresses to proliferative diabetic retinopathy (PDR) based on the presence of new blood vessels [37].

Depending on its form, DR has different fundus manifestations. In the NPDR stage, arteriolar hemangiomas, punctate retinal hemorrhage, hard exudation, cotton-wool spots, retinal edema, and beaded venous dilatation can be found. Angiogenesis is the main feature in the PDR stage with/without vitreous or preretinal hemorrhage [37]. Poor glycemic control leads to diabetic macular edema (DME), causing major visual impairment globally [34,37]. It can result in diffuse or focal areas of retinal thickening and/or exudates that lead to worsening visual acuity (VA), with half of patients losing two or more lines of vision within 2 years of DME [37,38].

Clinical trials such as the Diabetes Control and Complications Trial (DCCT) and the United Kingdom Prospective Diabetes Study (UKPDS), have shown that proper glycemic control can significantly reduce the risk of development and progression of DR [39,40]. Otherwise, chronic hyperglycemia can activate different metabolic pathways such as the polyol pathway and protein kinase C (PKC) activation pathway, leading to increased growth factors (VEGF) or insulin-like growth factor-1 (IGF-1) activity. These pathways lead to oxidative stress, causing severe retinal tissue damage [34,41,42].

The polyol pathway is a two-step metabolic pathway in which glucose is reduced to sorbitol, which is then converted to fructose. The aldose reductase enzyme (AR), which is localized in several retinal cells such as pericytes, endothelial cells, ganglion cells, Müller cells, retinal pigment epithelial cells and neurons, reduces glucose into sorbitol using nicotinamide adenine dinucleotide phosphate (NADPH) as a cofactor and sorbitol is converted into fructose by sorbitol dehydrogenase (SDH) [34]. It has been proven that AR’s increased activity and sorbitol accumulation (which is impermeable for cellular membranes) are responsible for homeostasis alteration and the evolution of glial and neuronal abnormalities as well as retinal cell destruction [43]. The ββ1/2 isoform of the protein kinase C (PKC) is highly associated with the DR development [34]. It has been proven that the synthesis of diacylglycerol (DAG), the key activator of PKC is increased in a diabetic/hyperglycemic state [43]. Being involved in pathophysiological processes such as inflammation, neovascularization, and hemodynamics, the activation of this enzyme influences retinal endothelial permeability, retinal hemodynamics, expression of vascular endothelial growth factor (VEGF), and leukostasis, which lead to DR progression [44].

VEGF, the growth factor most related to DR, is produced in ischemic and/or hypoxic situations. It is secreted primarily from retinal pigmented epithelial cells, pericytes, astrocytes, Müller cells, glial cells, and endothelial cells [44,45]. VEGF induces angiogenesis by interacting with angiotensin II, causing blood–retinal barrier (BRB) breakdown, endothelial cell injury, and retinal leucocyte adhesion by stimulating retinal intercellular adhesion molecule-1 (ICAM-1) and nitric oxide synthesis expression. Likewise, it increases vascular permeability via protein phosphorylation leading to DME. As a result, VEGF has been considered as a primary initiator of PDR, and as a potential mediator of NPDR [45,46].

When it comes to IGF-1, its role in DR progression is still unknown; however, increased levels of IGF-1 have been found in the vitreous fluid and serum of transgenic mice, which is believed to be linked to high glucose levels, VEGF, and other growth factors and subsequently DR development [45].

Recent studies showed that metabolites such as glutamic acid and aspartic acid were directly related to DR; however, because of the lack of accuracy, they cannot be used for the screening of DR nor the prediction of treatment response [35]. Glutamate is the major excitatory amino acid in the brain and retina and the dominant neurotransmitter of the retinal network, where glutamatergic synapses connect its fundamental functional cells, such as photoreceptors (PCs), bipolar cells (BCs), and retinal ganglion cells (RGCs) [46,47].

Müller cells maintain glutamate homeostasis by regulating its level both intra- and extra-cellularly within the retina. As a result, the disruption of glutamate homeostasis in the diabetic retina generates toxic levels of extracellular glutamate that may lead to neuron damage and ultimately initiate the development of DR [48]. The catabolism of BCAAs provides nitrogen for the synthesis of glutamate and glutamine using either of the two isoforms of branched-chain aminotransferase (BCAT): the mitochondrial branched-chain aminotransferase (BCATm) expressed in Müller cells or the cytosolic BCAT isoform (BCATc), which is expressed only in the cytosols of neuronal cells [48]. Therefore, it appears that increased ratios of BCKA/BCAA will decrease Müller cell glutamate synthesis and increase rates of glutamate oxidation, whereas decreasing the BCKA/BCAA ratios in diabetic retinas will promote glutamate excitotoxicity by raising the levels of glutamate in Müller cells. It is likely that a high serum BCAA level, which is typically found in diabetes, may lead to high intra- and extra-cellular glutamate levels in the retina, also interfering with glutamate clearance from the synaptic space and causing excitotoxicity to postsynaptic neurons, resulting in cell death and loss of vision [48].

Despite DR, diabetic patients are also known to be prone to diabetic papillopathy development. It is self-limiting and bilateral in half of the cases of disease and is characterized by optic disc swelling and axonal edema in and around the optic nerve head [49]. Also, cranial neuropathy may occur in diabetic patients, specifically III, IV, and VI nerve palsy, and sometimes the extraocular paresis can be the only sign of early-stage diabetes. In oculomotor nerve palsy, in a diabetic state, the pupillary reflexes are usually normal, differentiating it from an intracranial aneurysm or tumor [50].

Neovascularization of the iris (NVI), or rubeosis iridis, another DR complication, refers to the presence of fine blood vessels that appear on the anterior surface of the iris in response to retinal ischemia [38]. In advanced situations, NVI and neovascularization of the iridocorneal angle (NVA) can close the angle and lead to secondary angle closure glaucoma (NVG), which occurs in approximately 2% of diabetic patients and 21% of PDR cases [38]. DR is complex and intriguing; unfortunately, many patients are diagnosed in tardive stages, which is why an early screening with proper diabetic management could improve T2DM outcomes.

Regarding diabetic neuropathy, after extensive research, we could not identify any scientific articles that explored the relationship between strictly T2DM neuropathy and BCAAs. Given that hyperglycemia is induced via IR, which is responsible for the enhancement of the polyol pathway activity and sorbitol and oxidative stress accumulation, which are common pathways in DR, we may hypothesize to some extent that the link between diabetic neuropathy and BCAAs is similar to DR.

## 4. Diabetic Nephropathy

Diabetic nephropathy (DN) is a serious microvascular complication of T2DM and the major cause of chronic kidney disease [51]. It is also a major cause of morbidity and mortality in diabetic patients, as 30 to 40% develop DN [52]. DN leads to end-stage renal disease, requiring renal replacement therapy such as hemodialysis or kidney transplantation, which determines the necessity of its early detection [53]. DN is characterized by hyperfiltration and albuminuria in the early phases because of the extracellular matrix component accumulation and glomerular filtration barrier dysfunction [54], followed by a progressive renal function decline, which leads to end-stage kidney disease (ESKD) [55].

Transforming growth factor-β1 (TGF-β1) is considered both a fibrogenic and an inflammatory cytokine, and it plays an important role in the physiological functions and pathological states of the kidney [56]. High glucose levels can induce the expression of TGF-β1 in tubular and mesangial cells, accelerating the fibrosis of renal mesangial cells [52]. High glucose levels take part in the upregulation of TGF-β1 by inducing serine/threonine protein kinase/protein kinase B (Akt/PKB) phosphorylation in protein kinase C-β (PKC-β), which binds to a type II serine/threonine kinase receptor. Via this phosphorylation, it activates a type I receptor [56]. TGF-β1 binds to the TGF-β receptor II (T β RII), phosphorylates Smad2 and Smad3, and binds with Smad4 into a heterodimeric complex that regulates the transcription of TGF-β1 target genes, such as collagen a 1 (I), PAI- 1, Jun B, c -Jun, and fibronectin [57]. Bone morphogenetic protein-7 (BMP-7), a member of the TGF-β superfamily, can reduce glomerular and tubulointerstitial fibrosis and is considered to be a protective factor in DN [58]. BMP-7 activates the phosphorylation of Smad1/5/8 [52]. Phosphorylated Smad1/5/8 also binds the Smad4 protein and regulates the transcription of target genes. The activation of BMP-7-dependent Smad proteins inhibits the formation of TGF-β-dependent Smad2/3–Smad4 transcription factor complexes and the transcription of TGF-β-dependent genes [58]. Gremlin protein, a BMP-7 antagonist, is overexpressed in DN and is induced by high glucose levels [59]. It is considered a mediator of TGF-β and plays a distinct role in DN progression. These proteins are crucial in these processes and may hold therapeutic potential.

As we mentioned before, BCAAs are known as IR biomarkers and T2DM predictors. Higher intake of dietary BCAAs may increase the incidence of IR by more than 60% in adults and play an important role in the development of diabetes [60]. Moreover, short-term BCAA intake reduction seems to improve postprandial insulin sensitivity [61]. In DN, high serum BCAA levels are associated with a lower glomerular filtration and renal function decline [62]. In a study on 5/6-nephrectomized rats, the ones receiving a BCAA diet showed a decreased glomerular filtration rate and increased smooth muscle actin and collagen mRNA expression levels, suggesting renal dysfunction and kidney fibrosis [63]. Also, Zhenyukh et al. [64] showed the potentially harmful effect of BCAA on the vascular endothelium. Elevated BCAA levels induce inflammation and oxidative stress in endothelial cells by activating the mTORC1 pathway, which can lead to vascular hypercontractility and vasoconstriction. Complementary, chronic exposure to elevated BCAA levels may lead to atherosclerosis and other cardiovascular complications [54].

However, it was reported that in patients with renal failure, BCAA oral supplementation improves appetite and slows the progression of renal failure, counters oxidative stress in the kidneys, and alleviates diabetic kidney injury via the JNK/TGF-b/MMP-9 pathway [65]. BCAAs also protect renal mesangial cells from high-glucose-induced stress by attenuating the previously mentioned TGF-β1-Smad2/3 pathway and Gremlin expression and upregulating the BMP-7-Smad1/5/8 pathway [51]. These results should be cautiously interpreted, depending on the dose, the clinical context of the study (chronic or acute renal impairment), and other disease-related variables, if present.

The gut microbiota maintains internal homeostasis. Its dysbiosis promotes the production of bacteria-derived uremic toxins, such as indoxyl sulfate (IS), endotoxins, trimethylamine N-oxide (TMAO), and p-cresyl sulfate (PCS), which increase intestinal permeability and transfer into the systemic circulation. The accumulation of these toxins in the kidneys contributes to kidney dysfunction [66]. TMAO was found to be associated with high mortality, cardiovascular events, and poor renal outcomes in type 1 diabetes [67]. Phenyl sulfate (PS), one of the protein-bound uremic solutes, was proven to induce albuminuria and podocyte damage in experimental models of diabetes. In a diabetic patient cohort study, PS predicted a 2-year progression of albuminuria in patients with microalbuminuria [68]. There is an association between high levels of serum lipopolysaccharides (LPSs), impaired gut permeability, cardiovascular diseases, kidney disease (CKD), and patients with T2DM and CKD, as increased levels of circulating LPSs lead to excessive quantities of pro-inflammatory cytokine production and systemic inflammation (Figure 2) [69].

Systemic inflammation is a critical factor in the development of T2DM-related microvascular complications such as DR and DN. It is also difficult to treat these complications solely with antidiabetic medications, probably because the actual cell damage occurs before their diagnosis [70]. Consequently, for a better outcome, scientists are attempting to identify new biomarkers for disease prevention and novel adjuvant therapies in T2DM management and its associated macro- or micro-vascular complications.

## 5. Therapeutic and Biomarker Potential of BCAAs

As discussed, T2DM and its treatment represent a worldwide major public health burden [71]. In summary, T2DM is characterized by hyperglycemia, IR, or impaired insulin secretion, which leads to increased gluconeogenesis, glycogenolysis, and protein breakdown in the skeletal muscle. Elevated plasma concentrations of BCAAs have been found years prior to T2DM diagnosis, playing the role of predictive biomarkers for IR and T2DM but may also act as indicators of treatment effectiveness [38,72].

The most common management of T2DM includes lifestyle modifications with diet and exercise, and in cases where these are not sufficient, pharmacological treatment with antidiabetic drugs should be initiated. There are many antidiabetic agents, and their main goal is to reduce blood sugar levels and prevent the development of long-term complications such as diabetic nephropathy, neuropathy, or diabetic retinopathy [72].

Antidiabetic treatment is available as monotherapy or combination therapy, the latter involving at least two antidiabetic drugs and/or insulin. Antidiabetic agents include alpha-glucosidase inhibitors (acarbose, miglitol), amylin analogs (pramlintide), dipeptidyl peptidase-4 inhibitors (DPP-4) (alogliptan, linagliptan, saxagliptin, sitagliptin), incretin mimetics (GLP-1) (albiglutide, dulaglutide, exenatide, liraglutide, lixisenatide), meglitinides (nateglinide, repaglinide), biguanides (metformin), SGLT-2 inhibitors (canagliflozin, dapagliflozin, empagliflozin), sulfonylureas (chlorpropamide, glimepiride, glipizide, glyburide, tolazamide, tolbutamide), and thiazolidinediones (rosiglitazone, pioglitazone) [73]. Metformin is the drug of choice, as it increases insulin sensitivity and inhibits hepatic gluconeogenesis, reduces mortality and long-term complications, and promotes weight loss by lowering plasma lipid levels [72,73].

There are the following types of insulin: rapid-acting, which starts to work within a few minutes and covers insulin needs for meals eaten at the same time as the injection; regular- or short-acting, which works within 30–60 min; intermediate-acting, whose effects can last up to 18 h; long-acting, which can work for an entire day; and pre-mixed insulin, which is usually taken before every meal [74].

Some new therapeutic molecules are currently being investigated, such as GIP/GLP-1 Receptor Coagonist Tirzepatide, the first dual agonist approved for T2DM treatment, which has hypoglycemic weight loss effects and also presents great potential for cardiovascular protection [75].

As stated before, BCAAs that are positively correlated with insulin resistance and T2DM can be used as biomarkers in antidiabetic treatment [76]. Sriboonvorakul et al. [73] compared amino acid profiles for BCAAs, aromatic amino acids (AAA), and glutamate/glutamine in T2DM patients, which were divided into three groups. The first group followed monotherapy with metformin, and the second group took multiple drug therapies including metformin and sulfonylurea, while being compared to the healthy control group. The results showed that BCAA and tyrosine levels are reduced in patients with metformin or metformin-sulfonylurea combination therapies vs. healthy controls. These results emphasize the beneficial effects of antidiabetic oral molecules in amino acid control. Complementary, Bao et al. showed that rosiglitazone lowered BCAA levels following 6 months of therapy in T2DM patients [77], while Walford et al. [78] showed that BCAAs/AAAs decreased following glipizide treatment but increased in a metformin therapy group, with a stronger decrease in insulin sensitive than in insulin-resistant patients [79].

Empagliflozin (EMPA), an aSGLT-2 inhibitor drug known for its high selectivity for SGLT2, is the first-choice glucose-lowering agent that can significantly reduce cardiovascular risk in patients with T2DM [80] and may also help in diabetic retinopathy treatment [81,82]. Gong et al. showed that EMPA can lower BCAA levels in DR by boosting BCAA catabolism through an increase in BCAT levels and via BCKDK inhibition, which leads to BCKDH complex activation and may play a protective role in ameliorating the damaging effects of DR [83].

These findings show the potential use of BCAAs in investigating the mechanism of action of a certain drug, which may help in monitoring the responses to T2DM therapies, their effects, and their efficacy [73,83,84,85,86,87] (Table 1).

Recently, allostatic overload, a term used to describe the connection between environmental factors and the human genome, was associated with poor health outcomes such as elevated inflammatory responses, activation of the sympathetic nervous system, increased blood sugar levels, and IR [88]. It has an individual cumulative effect at both physiological and molecular levels, independent of exposure to social factors, nutrition, and lifestyle [89]. Nutrition is one of the key features that can directly influence microbial diversity, leading to inflammatory factors expansion, decreased methylation, and decreased antioxidant defenses. In that case, if BCAA catabolism is correlated to the amount and composition of the food, we may conclude that BCAA levels are dependable on external factors and human genome composure.

BCAA or BCKA supplementation are usually recommended to patients with chronic renal failure, who should follow a low-protein diet. BCAAs are also used as supplements for protein synthesis, recovery, and exercise-induced muscle damage [90]. However, it is still unclear if subjects with insulin resistance should be taking BCAA supplementation. Macotela et al. [91] showed that leucine improved glucose tolerance and decreased inflammation in adipose tissue in mice fed a high-fat diet, while Newgard et al. [92] showed that BCAA administration to rats on a high-fat diet increased insulin resistance.

Despite BCKA and BCAA levels being elevated in adults with obesity, it seems that BCKA levels are lower in adolescents with obesity than healthy-weight subjects [93]. Mccann et al. [94] suggest that there may be some changes in the regulation of BCAT and BCKDH during the transition from early adolescent to mature adolescent/adult adiposity.

Trub et al. [95] found that weight reduction led to an increased BCAA catabolism, and as a consequence, lower BCAA plasmatic levels and higher urea cycle amino acids were noted. It was also associated with insulin sensitivity improvement, assessed using triglycerides (TG)/high-density lipoprotein (HDL) and adiponectin levels. Weight reduction leads to a decrease in plasma glutamate, which may reduce long-term risks of cardiovascular disease and glucose intolerance [95].

After a weight loss program intervention, the participants showed an improvement in insulin resistance and lower BCAA plasmatic levels [96]. Two similar randomized dietary trials, i.e., POUNDS LOST and DIRECT, registered as NCT00072995 and NCT00160108, respectively, showed that both BCAAs and AAAs significantly decreased, with reduced metabolic risk seen in HOMA-IR values in response to weight-loss diet interventions.

A newly discovered therapeutic target for treating metabolic disorders secondary to BCAA accumulation is the enzyme branched-chain α-keto acid dehydrogenase kinase (BCKDK). BCKDK is an important enzyme that affects the metabolism of BCAA. In order to decrease BCAA levels, BCKDK should be inhibited [97]. In this regard, 3,6-dichlorobenzo[b]thiophene-2-carboxylic acid (BT2), a BCKDK inhibitor, showed remarkable results after 4 weeks of oral intake [98]. (s)-α-chloro-phenylpropionic acid ((S)-CPP) is also believed to have the same effect of inhibiting BCKDK levels and reducing BCAA concentrations [99]. Some studies have shown that pyridostigmine promotes BCAA catabolism by enhancing vagal activity and attenuating intestinal barrier injury and gut bacteria dysbiosis in diabetic cardiomyopathy mice. It can also upregulate BCAT2 and PP2Cm and downregulate p BCKDHA/BCKDHA and BCKDK in order to improve cardiac BCAA catabolism [100].

These data render the beneficial aspects in investigating BCAAs as potential predictive biomarkers or as therapeutic targets in T2DM and its complications.

## 6. Discussion

Although emerging experimental studies show the incremental position of BCAAs in metabolic diseases such as T2DM, there is a lack of larger human studies that explore the precise role of BCAAs, beneficial or not, as oral supplementation in T2DM and, particularly, in T2DM complications. From our knowledge, this is the first review that brings attention to the role of BCAAs in T2DM microvascular complications. It is worth mentioning that we directed our focus on diabetic retinopathy and diabetic nephropathy as there are few to no studies that investigated the possible connections between BCAAs and diabetic neuropathy.

Overall, the results showed higher levels of BCAAs in pre- or diabetic patients, and some suggested that the association between BCAAs and T2DM may be influenced by calorie intake [101]. Interestingly, the CARDIA study concluded that serial clinical metabolomic measurements in most young adults over a tracking period of 28 years identified subpopulations with rising levels of BCAAs, which was associated with high risk of DM in later life [79]. This gives us a new perspective on using BCAAs as biomarkers.

In this regard, metabolomic studies revealed that BCAAs and fatty acids (FAs) might be useful for monitoring the development of insulin resistance and thus, T2DM [102] and DR/DN [103].

Nevertheless, therapeutic lifestyle interventions reduce the risk of T2DM by reducing the circulating levels of BCAAs [104], and certified clinical trials are emerging that attempt to prove the potential usefulness of BCAAs as therapeutic agents and/or biomarkers in T2DM (Table 2).

## 7. Conclusions

As diabetes mellitus type 2 remains one of the most burdening diseases with multiple macro- and microvascular complications, there are continuous efforts to discover novel molecules for disease prevention and treatment. In particular, specific building blocks of the protein BCAAs including leucine, isoleucine, and valine are demonstrated to be associated with IR and diabetes development. However, depending on the presence of other clinical variables such as obesity, it is still unclear if subjects with IR should be taking BCAA supplementation. Thus, there is room for further research on their potential as biomarkers or as adjuvant therapeutic agents in T2DM and its related complications.

## Figures and Tables

**Figure 1 jcm-12-06053-f001:**
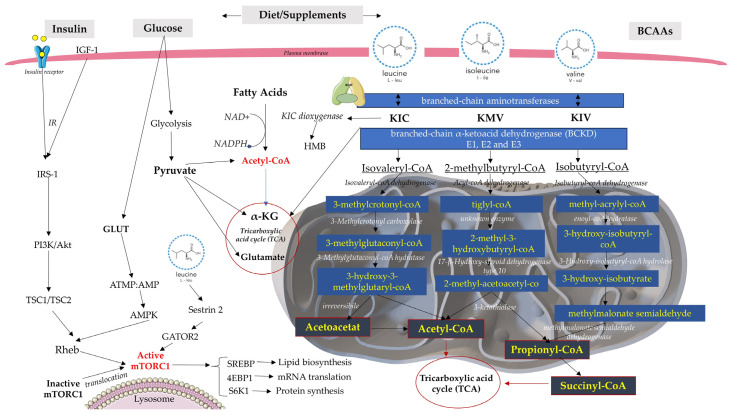
The BCAA catabolic and metabolic complex pathway regulates energy metabolism. Increased levels of BCAAs and fatty acids can interfere with normal insulin signaling through various mechanisms and ultimately lead to IR. In glycolysis and the Krebs cycle, NAD+ is reduced to form the NADH molecule. If NAD+ is not present, glycolysis will not be able to continue. IRS-1 (insulin receptor substrate-1); GLUT4 (glucose transporter type 4); SREBP (Sterol regulatory element binding proteins); Ras homolog enriched in brain (RHEB).

**Figure 2 jcm-12-06053-f002:**
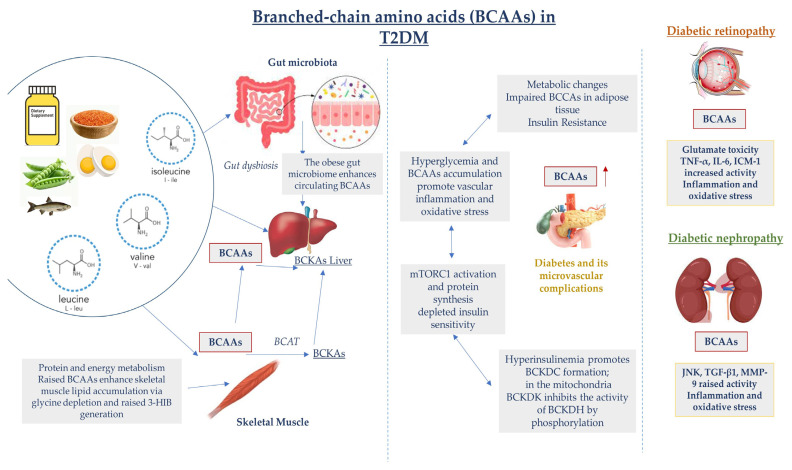
Dysregulated BCAA catabolism is highly associated with chronic inflammation and tissue damage in T2DM. Additionally, in the early stages of T2DM development, hyperglycemia and insulin resistance contribute to elevations in BCAA levels, which further promotes the activation of inflammatory pathways, oxidative stress with endothelial impairment, and vascular damage, leading to microvascular complication advancement. c-Jun N-terminal kinase (JNK); matrix metalloproteinase-9 (MMP-9).

**Table 1 jcm-12-06053-t001:** The effects of different antidiabetic drugs on BCAAs. Branched-chain α-keto acids (BCKAs); of mitochondrial branched-chain amino transferase (BCAT); non-alcoholic fatty liver (NAFL).

Source	Trial Type	Molecule	Outcomes
Sriboonvorakul et al. [71]	A clinical trial on a cohort of T2DM patients that was compared to healthy controls following treatment with single (metformin) or multiple drug (metformin and sulfonylurea).	Metformin and Sulfonylurea	In both of the treated T2DM groups, BCAAs were significantly lower than the healthy controls. Isoleucine was significantly lower in the single-treated T2DM group compared with the healthy controls. Valine was significantly lower in both treated T2DM groups compared with healthy controls. Leucine was significantly lower in both treated T2DM groups compared with healthy controls (*p* < 0.0001)
Gong et al. [83]	An experimental study on seven-week-old male diabetic *db*/*db* mice.	Empagliflozin	EMPA significantly inhibited oxidative stress and apoptosis and recovered tight junction in diabetic retinas. EMPA suppressed aberrant BCAA accumulation, which led to downregulation of inflammation and angiogenic factors, including TNF-α, IL-6, VCAM-1, and VEGF induced by diabetes. BCKAs were increased in diabetic retinas and decreased with EMPA application. BCKDK was enhanced and BCKDHA and BCKDHB were decreased in diabetic retinas
S Sonnet D et al. [84]	An experimental study on the iMSUD mouse model	Metformin	Metformin reduced levels of KIC in patient-derived fibroblasts by 20–50%; in the muscle by 69%, and in serum by 56% and restored levels of mitochondrial metabolites. Metformin decreased the expression of BCAT, which produces KIC in skeletal muscle.
Riviera et al. [85]	An experimental study on C2C12 mouse myoblasts (CRL-1772; ATCC, Manassas, VA).	Metformin	Metformin inhibited mitochondrial metabolism, promoting an activation of AMPK and subsequently PGC-1. Metformin reduced KLF15 (Kruppel-like factor 15) protein levels, leading to reduced expression/activation of BCAA catabolic enzymes. Metformin enhanced KLF15 mRNA expression, the implications of which were unknown.
Paterson et al. [86]	A clinical trial including nine non-insulin-dependent diabetic patients.	Gliclazide	Glycaemic control was improved, but fasting amino acid levels were not altered. Postprandial levels of BCAAs were significantly reduced: total BCAA (valine, leucine, and isoleucine) after 3 months of therapy (*p* < 0.01).
Iwasa M, et al. [87]	A clinical trial on 84 subjects with type 2 DM, NAFL, hypertension, and dyslipidemia.	Pioglitazone and Alogliptin	BCAA levels were negatively correlated with HDL cholesterol. BCAA levels were positively correlated with ALT, suggesting an association with fatty liver changes. There were no significant correlations with HbA1c, HOMA-IR, TG, hs-CRP, or adiponectin. Serum BCAA levels in diabetics were higher than in non-diabetics. Treatment with pioglitazone and alogliptin improved serum haemoglobin A1c and decreased BCAA levels.

**Table 2 jcm-12-06053-t002:** Clinical trials that investigated the preventive effects of dietary supplements with BCAAs in pre-diabetic, non-diabetic, and diabetic patients. Meal replacement beverages (MRBs); BCAD2 (branched-chain amino acid); hemoglobin A1c (HbA1c).

NCT Number Trial	Status	Molecule Investigated	Trial Type	Primary Outcomes
NCT02351323	Completed	glutamine and leucine	randomized, double-blind, placebo controlled, clinical trial	test the efficacy of 6 months of glutamine supplements in reducing biomarkers for IR and weight gain among 56 obese adolescents aged 12–19 years with a BMI ≥ 95th percentile and a family history of T2DM
NCT01211717	Completed	isoleucine, leucine, and valine	randomized, interventional clinical trial	determine the effectiveness of three BCAAs (isoleucine, leucine, and valine) on treating delayed onset muscle soreness in T2DM
NCT02435277 NCT02151461	Completed	leucine and metformin combinations	phase 2 trials randomized	change in fasting plasma glucose from baseline (day 1) to week 4 (day 28); change in HbA1c Levels
NCT01593605	Completed	resveratrol/leucine and resveratrol/HMB	randomized controlled trial	their ability to control glucose levels in persons without diabetes but with impaired fasting glucose
NCT04461236	Recruiting	isoleucine	randomized controlled clinical trial	change in whole-body protein metabolism in type 2 diabetic obese subjects; 24 h glucose levels in type 2 diabetic obese subjects
NCT04424537	Withdrawn	MRBs made with BCAD2 powder (lacking BCAAs)	randomized controlled trial	change in weight and fasting blood glucose level change in insulin sensitivity
NCT03785951	Unknown	wheat protein with leucine	double-blind, randomized, controlled, three-way, cross-over study	change in fasting and day-long glucose levels, and day-long insulin levels (using ELISA)

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
