# Peer review of "Depiction of Branched-Chain Amino Acids (BCAAs) in Diabetes with a Focus on Diabetic Microvascular Complications"

_jcm, 2023, doi:10.3390/jcm12186053_

Round 1

Reviewer 1 Report

This manuscript describes the link between BCAA and  diabetes chronic complications such as retinopathy and  nephropathy. However, the review needs extensive English language editing. In addition, I highly recommend adding a paragraph about changes in BCAA metabolic genes expression in diabetes.  

Figure one is well done, but you need to add main results in table one

The major issue about this review is English language. The topic and the scientific contents are good. However, There are some other comments that needs to be addressed such as:

English language needs extensive improvements thorough out the hall manuscript 

  • Some examples: 

Line 33-34: please consider rephrasing 

Line 39: remove “respectively”  

Line 40: “therapeutic concepts” what do you mean by this term? 

Line 50: Disfunsction should be dysfunction, the liver should be liver 

  • Gut microbiota paragraph is irrelevant to this review, and I suggest removing it 
  • Too many references, I suggest cutting it down to less than 100

Needs improvement

Author Response

Dear Reviewer,

Firstly, on behalf on our team we thank you for your time and all the recommendation made. Accordingly, we tried as much as possible to fallow them and correct our manuscript.

This manuscript describes the link between BCAA and diabetes chronic complications such as retinopathy and nephropathy. However, the review needs extensive English language editing.

The major issue about this review is English language. The topic and the scientific contents are good. However, There are some other comments that needs to be addressed such as: English language needs extensive improvements thorough out the hall manuscript

ï‚· Some examples:

Line 33-34: please consider rephrasing

We have rephrased this sentence as it was unclear.

Line 39: remove “respectively”

We have removed this word

Line 40: “therapeutic concepts” what do you mean by this term?

To avoid confusion, we replaced “concepts” with a more proper word.

Line 50: Disfunsction should be dysfunction, the liver should be liver

We have corrected.

We have checked the manuscript, and corrected all Grammer mistakes identified, also, we made structural changes.

In addition, I highly recommend adding a paragraph about changes in BCAA metabolic genes expression in diabetes.

We have introduced as suggested a paragraph about changes in BCAA metabolic genes expression in diabetes, “2.2. BCAAs metabolic genes expression in diabetes”, please verify.

Figure one is well done, but you need to add main results in table one.

Thank you for this remark, in table one now table two, we could not introduce results only the primary outcomes of the clinical trials involved, as even after rigorous research the studies have not published the results in official articles, one is still ongoing and one is Withdrawn; please verify.

Gut microbiota paragraph is irrelevant to this review, and I suggest removing it We have removed the paragraph about gut microbiota topic as suggested.

Too many references, I suggest cutting it down to less than 100

As recommended, after revision we have removed as much as possible references; however due to need of introducing new text, our bibliography is now at 107 references.

Reviewer 2 Report

 I congratulate the authors for the quality of the manuscript. The topic of the article is original and summarize well the subject.

There are no major flaws, and the overall structure is well done, with distinct paragraphs.

A few remarks:
- It would be interesting to diagram the pathophysiology of BCAAs catabolism and metabolism. The first paragraph is well written, but necessarily long. Any construction can be lightened by making a figure summarizing all the mechanisms described.

-What about diabetic neuropathy in its various forms? The authors have described retinal and nephropathic neuropathy. It would be interesting to include a paragraph on these conditions. Is there a relationship between the different types of neuropathy observed?

-  We can't help thinking about the worsening allostatic load in diabetic patients. By increasing insulin resistance, inflammation increases, along with oxidative stress factors. There is a direct relationship with the physiological aging of the body correlated with chronic hyperglycemia. The authors recalled the role of microbiota and diet.
It would be interesting to mention the interplay with allostatic load in this sense, whether related to microbiota (doi: 10.1042/BST20190110) or the Mediterranean diet (doi: 10.1016/j.psyneuen.2022.105841).

- A table resuming the different antidiabetics drugs and their intrication with BCAAs id very recommended to ensure more clarity in the therapeutic topic.

- The references are well documented and updated.

- The english is good.

Author Response

Dear Reviewer,

Firstly, on behalf on our team we thank you for your time and all the recommendation made. Accordingly, we revised and corrected all mistakes identified, introduced missing information and restructured the manuscript.

There are no major flaws, and the o verall structure is well done, with distinct paragraphs.

-It would be interesting to diagram the pathophysiology of BCAAs catabolism and metabolism. The first paragraph
is well written, but necessarily long. Any construction can be lightened by making a figure summarizing all the
mechanismsdescribed.

As recommended, we have revised and restructured the manuscript, and deleted unnecessary information and
included a complex figure which summariz es all the mechanisms described in the text.

-What about diabetic neuropathy in its various forms? The authors have described retinal and nephropathic
neuropathy. It would be interesting to include a paragraph on these conditions. Is there a relationship between the
different types of neuropathy observed?

As, co uld not identify scientific articles that explore the relationship between strictly T2DM neuropathy
and BCAAs , we have included a paragraph in which we give out opinion about this topic, inserted at the
end of DR chapter; as we believed to be more suitable ; please

-We can't help thinking about the worsening allostatic load in diabetic patients. By increasing insulin resistance,
inflammation increases, along with oxidative stress factors. There is a direct relationship with the physiological
aging of the body correlated with chronic hyperglycemia. The authors recalled the role of microbiota and diet.
It would be interesting to mention the interplay with allostatic load in this sense, whether related to microbiota (doi:
10.1042/BST20190110) or the Me diterranean diet (doi: 10.1016/j.psyneuen.2022.105841).

We have introduced a paragraph in which we bring to discussion about the allostatic load in diabetic
patients with transition to T2DM treatment and BCAASs involvement please verify.

-A table resuming the different antidiabetics drugs and their intrication with BCAAs id very recommended to
ensure more clarity in the therapeutic topic.

As suggested we included in our manuscript table 1, which resumes the different antidiabetics drugs and
their intrication with BCAAs . We hope that we respected your recommendation and improved our paper.

Reviewer 3 Report

The review titled "Depiction of Branched-Chain Amino Acids (BCAAs) in Diabetes: focus on Diabetic Microvascular Complications" is a well written paper. Authors reviewed the role of BCAAs in diabetes and its microvascular complications. The organisation of the text is fine. The subtitles are relevant. My unique suggestion is discussing inflammation's role as a link between BCAAs and diabetes mellitus. As far as we know that inflammation is associated with type 2 DM and related disorders including microvascular complication.

Author Response

Dear Reviewer,

Firstly, on behalf on our team we thank you for your time and all the recommendation made. Accordingly, we revised and corrected all mistakes identified, introduced missing information and restructured the manuscript.

-My unique suggestion is discussing inflammation's role as a link between BCAAs and diabetes mellitus. As far as
we know that inflammation is a ssociated with type 2 DM and related disorders including microvascular
complication.

We have introduced as suggested, a paragraph about the role of inflammation in BCAAs and diabetes
relationship, in chapter “ 2 .1. Insulin resistance and inflammation

Round 2

Reviewer 1 Report

No further comments

I am satisfied with the corrections made by authors